# Seaweeds as a "Palatable" Challenge between Innovation and Sustainability: A Systematic Review of Food Safety

**Giuseppe Cavallo** [1,*], **Chiara Lorini** [1], **Giuseppe Garamella** [2] **and Guglielmo Bonaccorsi** [1]

1. Department of Health Science, University of Florence, Viale GB Morgagni 48, 50134 Florence, Italy; chiara.lorini@unifi.it (C.L.); guglielmo.bonaccorsi@unifi.it (G.B.)
2. School of Specialization in Hygiene and Preventive Medicine, University of Florence, Viale GB Morgagni 48, 50134 Florence, Italy; dott.giuseppegaramella@gmail.com
* Correspondence: giuseppe.cavallo@unifi.it

**Abstract:** Moderate or severe food insecurity affect 2 billion people worldwide. The four pillars of food security (availability, access, use and stability) are in danger due to the impact of climatic and anthropogenic factors which impact on the food system. Novel foods, like seaweeds, have the potential to increase food yields so that to contribute in preventing or avoiding future global food shortages. The purpose of this systematic review was to assess microbiological, chemical, physical, and allergenic risks associated with seaweed consumption. Four research strings have been used to search for these risks. Preferred Reporting Item for Systematic Reviews and Meta-analysis (PRISMA) guidelines were applied. Finally, 39 articles met the selected criteria. No significant hazards for microbiological, allergenic, and physical risks were detected. Regarding chemical risk, algae can accumulate various heavy metals, especially when harvested in polluted sites. Cultivating seaweeds in a controlled environment allows to avoid this risk. Periodic checks will be necessary on the finished products to monitor heavy metals levels. Since the consumption of algae seems to be on the rise everywhere, it seems to be urgent that food control authorities establish the safety levels to which eating algae does not represent any risk for human health.

**Keywords:** novel food; food safety; seaweed; risk assessment; human health

## 1. Introduction

Moderate or severe food insecurity affects 2 billion people. The consequences of poor or insufficient intake of nutrients are malnutrition and, finally, poor health [1]. On the other hand, in 2016, the number of overweight adults was about 2 billion, of whom over one third were obese. In recent years, this scenario has spread from high-income countries to poor and developing countries [2]. The four pillars of food security (availability, access, use, and stability) are in danger due to the impact of climatic and anthropogenic factors on the food system [3]. Goal 2 of the 2030 Agenda for Sustainable Development highlights the importance of the link between the resilience of food production systems and the adaptation to climate change effects [4]. Looking ahead, the core question is whether the current agriculture and food systems will be able to satisfy the needs of a global population which could reach nearly ten billion people in 2050 with projections pointing to even greater growth by 2100 [5]. The food system is facing an unprecedented race to grab primary resources such as water, land and energy, increasing the negative impact on the environment, dramatically [6]. In the next future, food production will have to consider many aspects linked to sustainability, such as the reduction of waste, the earth's resources, the pollution, and consequently the chance to consume new foods to satisfy the growing demand, especially of proteic food [7]. Novel foods (NF), defined as "foods that had not been consumed to a significant degree by humans in the EU before 15 May 1997", when the first Regulation on novel foods came into force [8], represent a viable way to mitigate the impending shortage of food resources globally [9,10]. Previously, novel foods

were required mainly for their content of substances with beneficial actions on the body, such as omega 3 fatty acids or antioxidant agents. Recently, their use has been turning towards whole food, rather than extracts [11]. Among the novel foods, algae hide a great potential to meet the demand for macro and micronutrients, to forefront the growth of the population and the environmental impact [12,13]. Depending on the characteristics of the pigments, algae can be divided into brown (Phaeophyceae), green (Chlorophyceae), and red (Rhodophyceae), called also macro-algae. The height dimensions of brown algae range from 20 m to 30–60 cm. Red and green seaweeds vary their height from a few centimeters to a meter. The Rhodophyceae can also present themselves with shades of red, as brownish red and purple. The micro-algae (Cyanophyceae), also called blue-green algae, can be, frequently, unicellular with microscopic dimensions [14]. Each algal family has different species that are widely used for human food purposes. Some edible seaweeds are named differently from their latin name. For example, *Ascophyllum nodosum* (Knotted wrack), *Chondrus crispus* (Irish moss or Carrageen), *Himanthalia elongata* (Sea spaghetti), *Laminaria digitata* (Kombu), *Laminaria saccharina* (Royal or Sweet Kombu), *Porphyra umbilicalis* (nori), *Ulva lactuca* (Sea lettuce), *Undaria pinnatifida* (Wakame) [15]. Traditionally, the largest producers and consumers of algae are Asian countries, such as China, Japan, and South Korea. For a long time, Canada, USA and some European countries have used seaweeds as an ingredient. Globalization of food chains has favored export and consumption of algae as a food [16]. There are different ways of consuming algae as whole food: fresh, dried or cooked. Regarding the extracts, agar, alginates, and carrageenans, these have a long shelf life and are highly appreciated in food preparations. The use of algae in food production are many: we can find them in beverages, dairy products, meat, noodles, and soup [17,18]. Seaweed species contain a variable content of carbohydrates (ranging from 3 to 50%), proteins (from 7 to 75%), lipids (less than 5%), and ash (from 10 to 40%) [18,19]. Many factors influence their chemical and nutritional composition, both environmental (area of origin or cultivation, geographic position, seasons, environment temperature) and non-environmental (algal species, physiological variations, harvest time, and manufacturing process) [18,20]. Seaweed protein content ranges from 7 to 47 g/100 g of dry weight (DW) and it depends on the environmental factors mentioned above [21]. The algal proteins contain a complete profile of essential amino acids (EAA) and most of the EAA are available throughout the year although seasonal variations in their concentrations are known to occur [22]. Neutral lipids and glycolipids are the main components of the lipid pattern in algae. A very important nutritional characteristic consists of the content of long-chain polyunsaturated fatty acids such as eicosapentaenoic (EPA) and docosahexaenoic acids (DHA) [23]. Algae have, out of their total weight, a percentage of insoluble fibers from 39 to 93.5% and soluble from 6.5 to 61%, with alginates and carrageenans which have the function of storage polysaccharides [18,24]. Algal fiber content normally ranks from 32 to 75 g/100 g of dry weight [19,25]. The seaweeds vitamin content is high, especially vitamins B complex (B12 included) and C, and vitamins A and E [18,26].

Despite their nutritional potential and their raising consumption worldwide [27], some studies have reported potential risks for human health. Nonetheless, no specific regulations have been enacted for the use of algae as human food. The intakes have to conform to the general safety regulations for food and its contents, specified by a provisional TWI recommended by the World Health Organization based on an average adult body weight of 68 kg. Only France selected algae that can be introduced in the diet without risk to human health [16,28]. In Europe, the microbiological risk described in the Commission Regulation (EC) No 2073/2005 [29], foresees no specific criteria about seaweeds. Additionally, the chemical contamination of seaweeds for human consumption is not specifically regulated by law: currently no maximum levels (MLs) are established for seaweeds and halophytes, except for the MLs established for food supplements consisting exclusively or mainly of seaweeds or products derived from seaweeds [30]. The heavy metals limits were included in Commission Regulations 186/2015 EU [31], and 1275/2013 [32] for the animal feed, as well as being regulated by the French Centre d'Etude et de Valorisation des

Algues (CEVA) [33]. Regarding the allergenic risks, European Union (EU) has adopted the Regulation (EU) 1169/2011 [34], but seaweeds do not appear in annex II, which that specifies all the substances causing allergies and intolerances.

To the best of our knowledge, to date no systematic reviews have been published regarding the risks linked to human consumption of seaweeds. The aim of this study is to provide a systematic review on microbiological, allergenic, physical, and chemical risk associated with human consumption of seaweeds, to synthetize known risks and lack of information.

## 2. Materials and Methods

The Preferred Reporting Item for Systematic Reviews and Meta-analysis (PRISMA) statement was adopted to select the articles which satisfied the conditions below expressed. Four databases were explored: Pubmed, Embase, Scopus, and Web of Science; the time-frame considered is 1 January 2014 to 27 August 2020; the search strings for microbiological, allergenic, physical risk, and chemical risk were the following:

- Food AND (seaweed* OR "novel food*" OR alga*) AND (microbiota OR "microb* community" OR "microb* count*" OR "microb* load" OR "microb* risk" OR "microb* hazard" OR "microb* saf*" OR "food safety") for microbiological risk;
- Food AND (seaweed* OR "novel food*" OR alga*) AND (allergen* OR allerg*) for allergenic risk;
- Food AND (seaweed* OR "novel food*" OR alga*) AND ("physical risk*" OR "physical hazard*" OR "physical safety" OR "foreign bod*" OR "breeding substrate*") for physical risk;
- Food AND (seaweed* OR "novel food*" OR alga*) AND ("chemical risk*" OR "chemical hazard*" OR "chemical safety" OR radionuclide* OR metal* OR arsenic OR cadmium OR copper OR zinc OR chrome OR lead OR aluminium OR mercury OR toxin*) for chemical risk.

Only articles written in English, French, Italian, and Spanish were included. All duplicates were removed. For each type of risk investigated, a flow chart was produced to synthetize the results. Only primary studies published on peer-reviewed journals, aimed at assessing at least one of the investigated risks (microbiological, chemical or physical risk—including also allergenic risk) for human health were selected. Papers reporting the results of studies carried out on algal species to evaluate the contamination of microbiological, chemical or physical agents with well-known adverse effects for human health (including allergenic risk) were included as well. On the other hand, papers where the studies were conducted "in vitro", reporting only nutritional evaluation or investigating the risk for animals other than human were excluded. Reviews and the overview or similar, book chapters were excluded as well. The criteria were applied in each step of the selection process.

## 3. Results

A total of 5665 results were obtained: 2346 for microbiological, 898 for allergenic, 10 for physical, and 2411 for chemical risks. After screening by title, abstract, and full text, and removing duplicates, 39 articles were selected as the sum of the results of all the search strings (Figures 1–4).

Microbiological risk

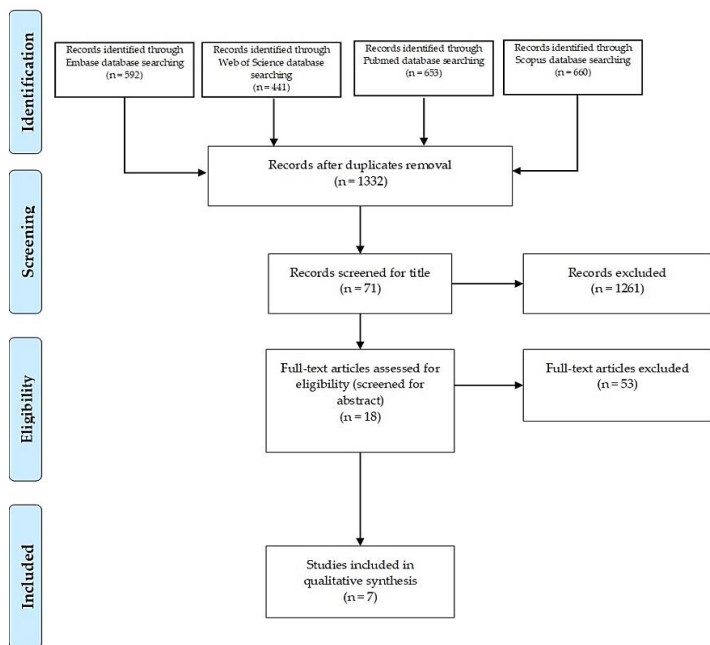

**Figure 1.** PRISMA (Preferred Reporting Items for Systematic reviews and Meta-Analyses) selection for microbiological risk. Full-text articles excluded were editorials, review, studies without clear methods and results, studies written in Chinese and German, oral communications, and congress abstracts.

Allergenic risk

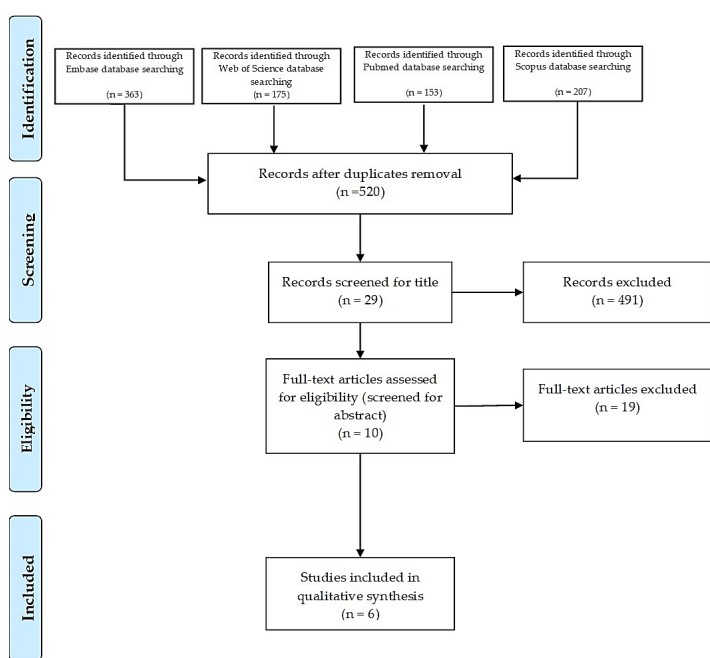

**Figure 2.** PRISMA (Preferred Reporting Items for Systematic reviews and Meta-Analyses) selection for allergenic risk. Full-text articles excluded were editorials, review, studies without clear methods and results, studies written in Chinese and German, oral communications, and congress abstracts.

Physical risk

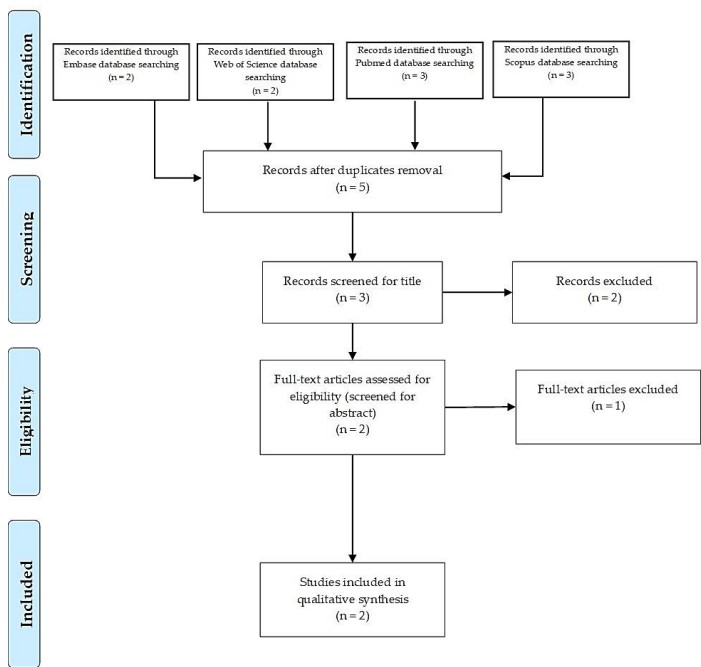

**Figure 3.** PRISMA (Preferred Reporting Items for Systematic reviews and Meta-Analyses) selection for physical risk. Full-text articles excluded were editorials, review, studies without clear methods and results, studies written in Chinese and German, oral communications, and congress abstracts.

Chemical risk

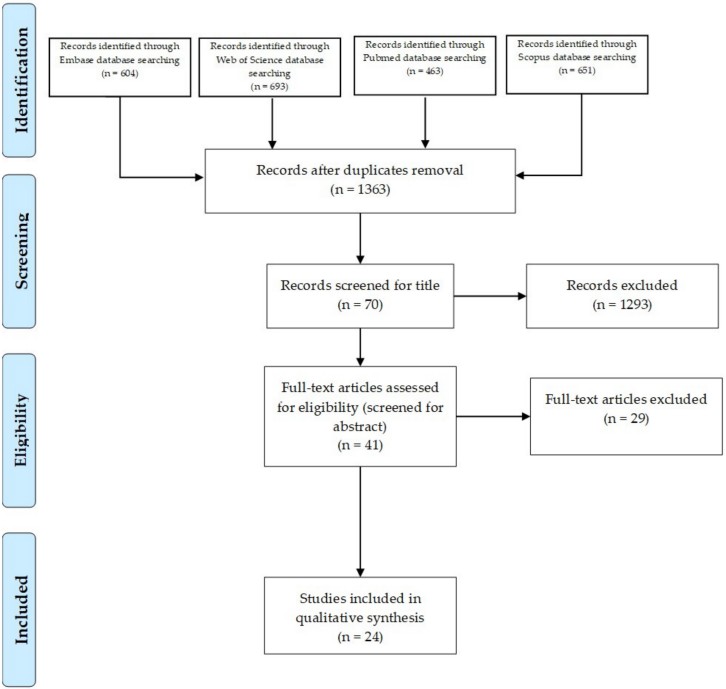

**Figure 4.** PRISMA (Preferred Reporting Items for Systematic reviews and Meta-Analyses) selection for chemical risk. Full-text articles excluded were editorials, review, studies without clear methods and results, studies written in Chinese and German, oral communications, and congress abstracts.

Of these, 33 were laboratory research studies and 6 were case reports. Regarding the countries where the studies were carried out, the articles assessing microbiological risk referred to Italy (2), Japan (1), Norway (1), South Korea (2), and the USA (1). As for the allergenic risk, three studies were conducted in Japan, and one in Canada, Germany, and the Netherlands. The physical risk was examined in articles referred to Italy and Japan. The chemical risk was investigated in research carried out in Brazil (1), Denmark (1), China (4), India (1), Indonesia (1), Italy (4), Japan (2), Kuwait (1), Norway (2), Poland (1), Portugal (1), Saudi Arabia (1), Spain (2), USA (2).

### 3.1. Microbiological Risk

Seven studies were included in the final review. The results were summarized in Table 1.

**Table 1.** Main characteristics of the studies included in the systematic review for the microbiological risk. Abbreviations: CFU: colony-forming unit; HT: Heat-treated; APC: aerobic plate count; ND: not detected; MPN: most probable number; RR: relative risk; CI: confidence interval.

| Author, Year | Country | Aim | Sample | Main Results |
|---|---|---|---|---|
| Blikra et al., 2018 | Norway | Among other goals, this study investigated the microbiological parameters of *Saccharina latissima* and *Alaria esculenta*. | *Alaria esculenta* and *Saccharina latissima*. | The results (expressed in log CFU/g) are reported, where possible, as mean $\pm$ SD. Total microbial count: *A. esculenta* 2.01 $\pm$ 0.39 (raw) and 1.20 $\pm$ 0.25 (Heat-treated, HT); *S. latissima* 1.10 $\pm$ 0.14 (raw) and 1.13 $\pm$ 0.18 (HT). Psychotropic bacteria: *A. esculenta* from 1 to 1.1 (raw) and 1 $\pm$ 0.01 (HT); *S. latissima* from 1.01 to 1.16 (raw), and 1 $\pm$ 0.01 (HT). Spore-forming bacteria aerobic: *A. esculenta* from 1.70 to 2.28 (raw) and from 1 to 2.9 (HT); *S. latissima* 1.01 $\pm$ 0.01 (raw) and from 1 to 2.9 (HT). Spore-forming bacteria anaerobic: *A. esculenta* from 1.47 to 1.7 (raw) and from 1 to 2.33 (HT); *S. latissima* from 1.01 to 1.7 (raw), and 1.08 $\pm$ 0.20 (HT). |
| Choi et al., 2014 | South Korea | The study analyzed microbiological profile from raw materials to final seasoned roasted laver products. | Korean laver. | The dried laver: APC level of 4.4 to 7.8 log CFU/g. Supplementary Materials: APC levels from not detected to 4.5 log CFU/g. Coliforms: 2.1 log CFU/g in dried laver and 1.8 log CFU/g in the primary roasting product. Microorganism species found: the main, with 9.7%, was *Moraxella* followed by *Clostridium* (no *botulinum* and *perfrigens*) with 8.4%, *Staphylococcus* (not *aureus*) with 8.1%, *Bacillus* (not *cereus*) with 6.6%, but not *B. cereus*), and *Neisseria* with 5.3%. The microbial populations during processing decreased from 6.9 log CFU/g in the dried laver to >3.2 log CFU/g in the packaged product. Only *B. cereus*, among pathogenic bacteria, was detected during production process. |

**Table 1.** *Cont.*

| Author, Year | Country | Aim | Sample | Main Results |
|---|---|---|---|---|
| EFSA NDA Panel, 2020 | Italy | The study expressed a scientific opinion about the risk associated with the consumption of dried whole cell *Euglena gracilis* as a novel food (NF) pursuant to Regulation (EU) 2015/2283. | *Euglena gracilis*. | The Aerobic plate count (CFU/g) for each of five batches ranged from 1400 to 8000. Coliforms (most probable number, MPN/g) were <3 in all batches. Yeast and mould (CFU/g) detected varied from 20 to 70. *Escherichia coli* (in 10 g) was absent in all batches, as well as *Staphylococcus aureus* (in 10 g), *Salmonella* (in 25 g), and *Listeria monocytogenes* (in 25 g). |
| EFSA NDA Panel, 2017 | Italy | The study expressed a scientific opinion on a dietary supplement composed of *Ecklonia cava* phlorotannins. | Phlorotannin-rich alcohol extract of *E. cava*. | The viable cell count was <3000 CFU/g. Moulds and yeasts were <300 CFU/g. *Ecklonia cava* samples were negative for *Staphylococcus aureus* (CFU/g), *Salmonella* ssp. (CFU/25 g), and Coliforms (CFU/g). |
| Nayyar et al., 2018 | USA | The study observed, among various parameters, microbial count of two seaweeds at 2 °C and 7 °C storage temperatures. | *P. palmata* and *G. tikvahiae*. | *P. palmata* microbial count: at 2 °C, it started from 4 CFU/g to less than 5 log CFU/g after 11 days of storage; at 7 °C, the growth range was 3–4 CFU/g. *G. tikvahiae* microbial count: at 2 °C, it started from 4 to 8 CFU/g, after 12 day of storage; at 7 °C, the bacterial growth performed from 4 to 7 CFU/g at the end of storage. |
| Park et al., 2014 | South Korea | The study presented the norovirus outbreaks linked to green alga consumption. | Number of cases: ninety-one students. | The symptoms that occurred most frequently were vomit, in all cases, followed by abdominal pain and nausea. The trigger for the symptoms was the ingestion of seasoned green algae which is therefore the cause of the norovirus outbreak |
| Sakon et al., 2018 | Japan | The study reported an epidemiological investigation on norovirus outbreaks associated with nori consumption. | 2094 consumers of contaminated nori. | Gastrointestinal symptoms were reported from five schools, a commercial office, and a shop. Investigations led to the conclusion that the responsible agent was the norovirus contained in shredded nori. The possible origin of the outbreaks was identified in the different processing stages of the various manufacturing companies. |

Blikra et al. [35] reported a low bacterial load on selected algae. In particular, no coliforms, enterococci, Listeria monocytogenes, or pathogenic vibrio were observed. About Bacillus pumilus and B. licheniformis, particular attention must be paid during the production process. Choi et al. [36] examined the Aerobic Plate Count (APC) during the entire processing cycle of the seasoned roasted laver. A reduction of the microbial population from raw material to finished product was observed. Only B. cereus was detected among pathogenic species for humans. EFSA [37] registered a lower aerobic plate count than the EFSA specification for Novel Food, as well as for coliforms, and yeast and moulds in Euglena gracilis. No other foodborne pathogenic bacteria were identified. In another EFSA technical report [38] three samples of Ecklonia cava for the presence of Salmonella spp.

were analyzed. No coliforms, yeast and moulds, Salmonella, and Staphylococcus aureus were detected. Nayyar et al. [39] investigated the effects of two storage temperatures on the microbial changes of two fresh red seaweeds. The microbial counts of G. tikvahiae were observed to increase during storage, contrary to what happened with P. palmata. An article of Park et al. [40] reported an outbreak of gastroenteritis occurred in two schools. A retrospective cohort study conducted in a school showed an association between symptoms and the raw green algae consumption. A case–control study in the other school highlighted the association between cases and the consumption of seasoned green algae with pears. Norovirus was identified in the samples analyzed. These outbreaks were correlated to the green seaweeds consumption. Sakon et al. [41] reported seven foodborne norovirus outbreaks in four Japanese areas that caused sickness in more than two-thousands persons. Nori consumption was associated with the outbreaks.

### 3.2. Allergenic Risk

Six studies were included in the final synthesis. The results were summarized in Table 2.

**Table 2.** Main characteristics of the studies included in the systematic review for the allergenic risk. Abbreviations: C-PC: phycocyanin C; NS: nori sauces.

| Author, Year | Country | Aim | Sample | Main Results |
|---|---|---|---|---|
| Kular et al., 2018 | Canada | The study described a case study, represented by an allergic reaction to the carrageenan. | A 10-month-old male with lip angioedema. | Following the consumption of a fruit cake, angioedema appeared on the lips. The skin prick test was performed which revealed the allergic reaction to carrageenan. |
| Lang-Yona et al., 2018 | Germany | The study analyzed the allergic reactivity of some cyanobacteria. | *Anabaena ambigua, Cylindrospermum siamensis, Lyngbya lagerheimii, Microcystis aeruginosa, Nostoc* sp., *Phormidium* sp., *Planktothrix agardhii,* and *Synechocystis* sp. | After laboratory analysis, C-phycocyanin (a pigment-protein complex, characteristic of the *Spirulina* algae) emerged as promoter of cross-reactivity with fresh marine species, compared to the other algae considered in the study. |
| Le et al., 2014 | Netherlands | The study described a case of allergic reaction after consuming *Spirulina* tablet. | 17-year-old male who showed symptoms associated with an allergic reaction after consuming a *Spirulina* food supplement. | The skin prick test confirmed the association. Phycocyanin was identified as potential allergenic element. |
| Thomas et al., 2018 | Japan | The study described a case of food allergy after consuming seaweed. | A 27-year-old man with a possible seafood allergy. | Prick to prick test for nori gave a positive result. Further investigations on red seaweeds, revealed positivity to the other species of the same algal family. Prick to prick tests for green and brown seaweeds were negative. |
| Uchida et al., 2017 | Japan | The study investigated the possible allergenic potential of nori sauce, and the cross-reactivity with other possible allergenic food. | Nori sauces set in different batches. | Allergens detected: wheat, soy, crustaceans (shrimp and crab). Results: all were negative (<0.1 mg/100 g). |
| Uchida et al., 2018 | Japan | The study investigated the allergenic potential of low-quality nori (LNs) sauce, and the cross-reactivity with other possible allergenic food. | Low-quality nori from Japan. | Allergens detected: wheat, soy, crustaceans (shrimp and crab) for low-quality nori and high-quality nori sauces. All tests were negative. |

Kular et al. [42] presented a singular case of carrageenan allergy in an infant with a lip angioedema after fruitcake icing ingestion. A skin prick test (SPT) confirmed the allergy to the carrageenan as ingredient of the pie. Lang-Yona et al. [43] observed the main allergenic role of C-phycocyanin (C-PC) as component of Spirulina. Le et al. [44] presented a case of an anaphylactic reaction following ingestion of Spirulina tablet. A 17-year-old male, ten minutes after the meal, presented abdominal pain, angioedema, dyspnea, nausea, tingling

of the lips, and urticaria of arms and trunk. The SPT with Spirulina revealed a positive reaction. Thomas et al. [45] reported a case of a 27-year-old man with a possible seafood allergy that caused him urticaria with facial angioedema. Prick to prick test revealed a positivity to the fresh nori alga and other two red alga species. Uchida et al. [46] reported that nori sauces (NSs) did not present cross-allergenicity with some seafood, wheat flour and soybean. The same result [47] was obtained for the low-quality nori sauce (LNS), also.

### 3.3. Physical Risk

Two articles were included in the final synthesis. The results are summarized in Table 3.

**Table 3.** Main characteristics of the studies included in the systematic review for the physical risk. Abbreviations: PCF: pharyngocutaneous fistula.

| Author, Year | Country | Aim | Sample | Main Results |
|---|---|---|---|---|
| Kusaba et al., 2019 | Japan | The study described a case of pharyngocutaneous fistula (PCF) following Kombu ingestion. | 63-year-old person who ingested Kombu meal. | A choking episode occurred after a seaweed meal with dried Kombu. The patient underwent an emergency surgery, consisting of drainage of the abscess and the removal of a foreign body. Videoscopy revealed the presence of alga consumed in the pharyngo-esophageal tract. |
| Panebianco et al., 2019 | Italy | The study assessed chemical, microbiological, and physical hazards related to seafood commercial products. | Twenty-six differently processed seaweeds samples. | During an inspection of some seafood products, three triangular glass bodies (identified by Scanning Electron Microscopy) were found in processed nori sample. |

Kusaba et al. [48] presented the case of a Japanese adult man in whom a foreign body was revealed in the digestive tract by video endoscopy, after dried Kombu consumption. Panebianco et al. [49] detected, with X-ray analysis, the presence of a triangular structure, probably from material used during seaweed processing.

### 3.4. Chemical Risk

Twenty-four studies were included in the final synthesis. The results were summarized in Table 4.

**Table 4.** Main characteristics of the studies included in the systematic review for the chemical risk. Abbreviations: The concentrations were reported as mean ± standard deviation. sp.: species (singular); spp.: species (plural); CF: concentration factors; HI: hazard index; DW: dry weight; LOQ: limit of quantification; SW: seaweeds; BW: body weight; PTMI: provisional tolerable monthly intake; FDA: Food and Drug Administration; PAHs: Polycyclic Aromatic Hydrocarbons; THQ: target hazard quotient; REEs: Rare Earths Elements; MAC: maximum acceptable concentration; LREEs: light Rare Earths Elements; iAs: inorganic arsenic; EU-FORA: The EUropean FOod Risk Assessment; fdw: freeze dried weight.

| Author, Year | Country | Aim | Sample | Main Results |
|---|---|---|---|---|
| Ali et al., 2019 | Saudi Arabia | The aim of this study was to detect the heavy metals levels in some red seaweeds, evaluating also the risk associated with their consumption for children and adults. | *Corallma, Gracilaria, Hypnea, Jania*, and *Laurencia* spp. | Heavy metal concentrations (µg/g) in red algae samples reported: Cr 11.7.86, Ni 6 ± 1.78, Cu 5.9 ± 2.93, Cd 0.09 ± 0.02 Pb 1.5 ± 0.41. The HI values did not represent a danger to the risk of cancer in adults and children. |

**Table 4.** *Cont.*

| Author, Year | Country | Aim | Sample | Main Results |
|---|---|---|---|---|
| Ardiyansyah et al., 2019 | Indonesia | This study reported data on Cd content in red alga species. | *Gracilaria* sp. | Cd concentration in seaweed samples: <0.0024 mg/kg. |
| Arulkumar et al., 2019 | India | The study established the concentration of Cd, Cu, Pb, and Zn in Chlorophyta, Ochrophyta, and Rhodophyta. | Thirteen edible seaweed. | Cd concentration: from $0.58 \pm 0.13$ to $5.24 \pm 0.99$ mg/kg. Pb concentration: from $14.20 \pm 0.87$ to $17.33 \pm 0.9$ mg/kg. Cu concentration: from $0.87 \pm 0.07$ to $8.62 \pm 0.77$ mg/kg. Zn concentration: from $19.59 \pm 0.63$ to $23.45 \pm 1.03$ mg/kg. |
| Biancarosa et al., 2017 | Norway | The study assessed the heavy metals and metalloids concentration in brown, green and red algae. | Twenty-one species of marine seaweeds. | Cd content: in green algae 0.12–0.18 mg/kg DW; in red algae 0.07–3.1; in brown algae 0.03–2.6 mg/kg DW. Hg concentration: from <LOQ to 0.04 mg/kg DW. Pb concentration: up to 0.58 mg/kg DW in red and brown algae; up to 3 mg/kg DW in green algae. As content: 21–120 mg/kg DW in brown algae, 6.4–24 mg/kg DW in red algae, 6.4–10 mg/kg DW in green algae. iAs concentration: <0.5 mg/kg in all samples. |
| Chen et al., 2018 | China | The study aimed to determine the metal content of algae, evaluating the risk linked to their consumption. | About three hundred Chinese seaweeds. | Concentration found in red seaweeds (mg/kg): Al $597.6 \pm 594.23$; As $22.05 \pm 11.28$; Cd $2.225 \pm 1.23$; Cr $2.545 \pm 4.08$; Cu $11.049 \pm 6.277$; Hg $0.01 \pm 0.017$; Ni $1.642 \pm 1.211$; Pb $0.655 \pm 0.474$. Concentration found in brown seaweeds (mg/kg): Al $597.65 + 655.65$; As $23.01 \pm 15.67$; Cd $0.245 \pm 0.286$; Cr $2.465 \pm 4.277$; Cu $2.33 \pm 5.468$; Hg $0.055 \pm 0.0619$; Ni $1.123 \pm 1.219$; Pb $0.539 \pm 0.63$. |
| Filippini et al., 2020 | Italy | The study explored heavy metal concentration in seaweeds, and health risk for adults and children associated with their intake. | Brown, green, mixed, and red seaweeds. | Al levels found: from 0.71 mg/kg to 165.39 mg/kg. Cd concentration: from 0.02 to 1.56 mg/kg). Pb concentration: from 0.16 to 0.56 mg/kg. Iodine content: from 10.66 to 6670.8 mg/kg. Hg level: <0.03 mg/kg. |
| Francisco et al., 2018 | Portugal | One of the objectives in this work was to assess the risk linked to the heavy metal in a brown alga species. | *Fucus spiralis*. | As concentration: $24.36 \pm 2.04$ µg/g DW. Cd content: from 0.07 (unpolluted site) to up to 3.58 µg/g DW (polluted site). I levels: $190 \pm 18$ µg/g DW. Hg and Pb values: low. |
| Kim et al., 2019 | USA | The study measured heavy metal content in a brown alga and in a red alga. | *Gracilaria tikvahiae* and *Saccharina latissima*. | *Gracilaria tikvahiae*: As, Cd, and Hg high levels in western Long Island Sound; Pb high in Bronx River Estuary. *Saccharina latissima*: Cd and Pb high levels in Bronx River Estuary. |
| Li et al., 2018 | China | The study evaluated the contaminants content in *U. prolifera* and the risks for human health associated with its consumption. | Samples of *U. prolifera*. | Heavy metal mean concentration of two years: As 0.66–0.93, Cd 0.0068–1.6, Cr1.6–9.7, Cu 1.7–10, Pb 0.064–2.3 mg/g DW. THQ: $<10^{-1}$. Potential carcinogenic PAHs: from 0.19 to 414 ng/g DW. |

**Table 4.** *Cont.*

| Author, Year | Country | Aim | Sample | Main Results |
|---|---|---|---|---|
| Liu et al., 2017 | China | This study investigated *Porphyra* REEs content and health risk related to its consumption. | Thirty-six *Porphyra* seaweed samples from Jiangsu province and Fujian province. | REEs content: from 2.187 to 13.452 mg/kg. Ce, La, Nd, and Y values were high in every sample. |
| Miller et al., 2020 | USA | The study analyzed the presence of anabaenopeptins, cyanopeptolins, and microginins, as well as microcystins (MC) variants in algal dietary supplements. | A total of 18 algal dietary supplement products containing cyanobacterial species *Aphanizomenon flos-aquae* (AFA) in pill form, whether capsule or tablet. | The analysis showed that the MAC was exceeded by forty to sixty times. Almost a quarter of the samples analysed presented an excess of MC. |
| Mise et al., 2019 | Japan | The study examined the dietary exposure to As in Japanese pregnant women and children for *Hijiki* seaweed consumption. | 104 pregnant women and 106 children. | Total arsenic intake: 8.46 µg/kg BW/week in pregnant women; 20.07 µg/kg BW/week in children. iAs values: 1.74 µg/kg BW/week for pregnant women and 4.81 µg/kg BW/week for children. |
| Panebianco et al., 2019 | Italy | The study assessed the presence of microbiological, chemical, and physical hazards in seaweeds. | Two seaweeds among seafood commercial products considered in the study. | For lead, the presence was 14% max in one sample and 17.8% max in the other. For As, the values were 4.1% max in the first sample and 6.7% max in the second. |
| Paz et al., 2018 | Spain | The goal was to assess the content of Al, Cd, and Pb and B, Ba, Fe, Li, Ni, and V, in several European seaweeds. | Sixty-four Phaeophyta species. | The following concentrations were expressed in mg/kg DW. Al in *Halopteris scoparia* $161 \pm 15.6$, in *Padina pavonica* $256 \pm 179$, in *Sargassum fluitans* $57.7 \pm 15.3$, in *Cystoseira* spp. $145 \pm 147$, in *Haliptilum virgatum* $36.3 \pm 4.02$. Cd in *H. scoparia* $0.07 \pm 0.01$, in *P. pavonica* $0.20 \pm 0.21$, in *S. fluitans* $0.16 \pm 0.06$, in *Cystoseira* spp. $0.19 \pm 0.09$, *H. virgatum* $0.19 \pm 0.02$. Ni in *H. scoparia* $1.54 \pm 0.4$, in *P. pavonica* $3.7 \pm 3.02$, in *S. fluitans* $0.9 \pm 0.34$, in *Cystoseira* spp. $1.6 \pm 0.64$, in *H. virgatum* $1.34 \pm 0.17$. Pb in *H. scoparia* $3.1 \pm 0.67$, in *P. pavonica* $3.92 \pm 3.71$, in *S. fluitans* $0.4 \pm 0.22$, in *Cystoseira* spp. $1.1 \pm 1.31$, *H. virgatum* $0.31 \pm 0.06$. |
| Paz et al., 2018 | Spain | The goal was to define the content of Al, Cd, Hg, and Pb in some edible seaweeds. | Seventy-three European and Asian seaweeds: *Undaria pinnatifida, Himanthalia elongate, Laminaria ochroleuca,* seaweed salad. | Al concentration: from a minimum of $19.1 \pm 8.6$ (Europe) to a maximum of $57.7 \pm 35$ mg/kg (Asia). Cd concentration: from $0.04 \pm 0.03$ to 1.11 (Europe) $\pm 0.3$ mg/kg (Asia). Hg levels: from <LOQ (Asia) to $0.024 \pm 0.001$ (Europe). Pb values: from $0.23 \pm 0.07$ (Europe) to $0.49 \pm 0.2$ (Asia). |

**Table 4.** *Cont.*

| Author, Year | Country | Aim | Sample | Main Results |
|---|---|---|---|---|
| Rzymski et al., 2019 | Poland | The study determined Al, As, Cd, Cr, Hg, Ni, Pb, and REEs content in *Spirulina* and *Chlorella* food supplements. | *Spirulina* and *Chlorella* supplements purchased online. | Al content 2155.6 ± 1774.7 (1299.8) mg/kg in *Spirulina*; 1732.8 ± 1991.5 mg/kg in *Chlorella*. Cd concentration: 0.125 ± 0.055 mg/kg in *Spirulina*; 0.142 ± 0.071 mg/kg in *Chlorella*. Cr (VI) content: below the detection limit (0.01 mg/kg) in both species. Hg values: 0.027 ± 0.031 mg/kg in *Spirulina*; 0.41 ± 0.017 mg/kg in *Chlorella*. iAs levels: 1.7–2.2 mg/kg in *Spirulina*; 2.3–2.7 mg/kg in *Chlorella*. Ni concentration: 1.52 ± 0.72 mg/kg in *Spirulina*; 1.38 ± 0.63 mg/kg in *Chlorella*. REEs amount: 2.14 ± 1.89 mg/kg in *Spirulina*; 2.03 ± 11.28 mg/kg in *Chlorella*. Pb content: 2.6 ± 1.9 mg/kg in *Spirulina*; 2.6 ± 1.3 mg/kg in *Chlorella*. LREEs and Cu levels: high in both species. |
| Sa Monteiro et al., 2019 | Denmark | The main goal was to determine the levels of iAs, Cd, Hg, I, and Pb in edible seaweeds. | Species selected for this study: *Fucus vesiculosus*, *Fucus serratus*, *Fucus spiralis*, *Fucus evanescens*, *Saccharina latissima*, *Ulva lactuca* and *Cladophora* sp. | As content: from 3.2 to 116.7 µg/g fdw. Cd values: 0.017–1.97 µg/g fdw. Hg concentration: from 0.003 to 0.042 µg/g fdw. I content: from 17.2 to 4782 µg/g fdw. iAs content: not available. Pb levels: from 0.072 to 9.6 µg/g fdw. |
| Santos-Silva et al., 2018 | Brazil | This study defined, in Phaeophyta and Rhodophyta, the background levels of As, Cd, Cu, Hg, and Pb. | Species selected for this study: *Dictyopteris delicatula*, *Canistrocarpus cervicornis*, *Ceratodictyon variabile* and *Palisada perforata*. | As content: from <4.84 to 23.21 µg/g. The maximum As values, from 13.46 µg/g to 49.52 µg/g, were detected in brown seaweeds. Cu concentration: from below the detection limit to 33.55 µg/g. Cd, Hg, and Pb: below the detection limits. Hg concentration: 22.25 µg/g, maximum value. The highest mean value of Cd was found in *Dictyopteris delicatula* (0.18 µg/g). |
| Stévant et al., 2018 | Norway | The aim of this work was to assess toxicity due to some heavy metals in cultivated *A. esculenta* and *S. latissima*. | Samples of *A. esculenta* and *S. latissima* from the Northern coast of France. | Initial content of the Cd, I and iAs, expressed in mg/kg DW. *A. esculenta* Cd concentration 2.01 ± 0.09 and 1.55 ± 0.2. *S. latissima* Cd concentration 0.22 ± 0.03 and 0.27 ± 0.01. Limit value: 0.5. *A. esculenta* I concentration 213 ± 12; *S. latissima* I concentration 4898 ± 166 and 6568 ± 398. Limit value: 2000. *S. latissima* iAs content: 0.22 ± 0.04; *S. latissima* iAs content: 0.16 ± 0.02 and 0.23 ± 0.01. Limit value: 3. |
| Squadrone et al., 2018 | Italy | The study determined the levels of non-essential trace elements, essential trace elements, and risk for humans in several Mediterranean seaweeds. | Brown, green, and red algae species. | Main results: Brown algae: Al 9916 mg/kg, Pb 40 mg/kg. Green algae: As 37 mg/kg, Cd 0.32 mg/kg, Co 5.6 mg/kg, Cu 73 mg/kg. Brown algae contained the largest amount of total metals (20.172 mg/kg DW) the red algae the lower (8292 mg/kg DW). |

**Table 4.** *Cont.*

| Author, Year | Country | Aim | Sample | Main Results |
|---|---|---|---|---|
| Squadrone et al., 2019 | Italy | The study analyzed the REEs, among others, in seaweeds. | Brown, green, and red algae. | ∑REE concentration in seaweed: 12 mg/kg. |
| Uchida et al., 2016 | Japan | The present study attempted to measure the heavy metal contents of the NSs for food safety. | Dried sheets of Japanese nori alga. | Cd: 0.05 mg/100 g or lower. Total As: 0.8 mg/100 g iAs: <0.05 mg/100 g Cr: 0.01 mg/100 g or lower. Histamine: from 0.9 to 9.0 mg/100 mL. |
| Uddin et al., 2019 | Kuwait | The study registered $^{210}$Po and $^{210}$Pb and radionuclides concentrations in different algae species. | *Cladophora Sargassum*, and *Ulva.* | $^{210}$Po/$^{210}$Pb ratio: from 2.67 to 10.95. |
| Zhang et al., 2020 | China | The study delivered information about health risk associated with selenium-rice agro-food consumption. | Several thousands of selenium-rich agro-food purchased at the market. | Concentration of heavy metals in edible fungi and algae (µg/g wet weight): Pb content 0.3253/0.4580; As content 0.1947/0.3681; Cd content 0.2635/1.6763; Hg content 0.0086/0.1960. |

Ali et al. [50] produced a risk assessment for children and adult from intake of some algae species including metals in the Red Sea of the Indian Ocean. The sampling sites are dense with human activity, from industry to freight transport. No cancer risk emerged for children and adults. This study investigated the bioaccumulation of Cd, Cr, Cu, Mn, Ni, Pb, and Zn in red algae species from the sites considered. Significant variations of metal bioaccumulation levels among different species were observed. Ardiyansyah et al. [51] conducted a research on the bioaccumulation of cadmium on *Gracilaria* sp. in different sites of an Indonesian coastal district, an area where industrial and domestic waste was disposed. Results of the analysis of cadmium content in *Gracilaria* sp. from every reference station showed the same amount, below the threshold established by the National Standardization Agency of Indonesia (0.2 mg/kg). The cadmium bioaccumulation was found to be low in *Gracilaria* sp.. Arulkumar et al. [52] monitored the Cd, Cu, Pb, and Zn concentration in 13 edible brown, red, and green algal species collected from southeast India, in an ecosystem characterized by solid and liquid untreated municipal waste. Compared to brown and red algae, green seaweeds appeared to have a greater capacity to accumulate toxic metal. Biancarosa et al. [53] analyzed the heavy metals concentrations in 21 species of marine seaweeds. Cd, Hg, and Pb were detected in every alga considered. Green algae presented a lower concentration of Cd than red and brown algae. Hg levels in the species studied were low. Pb concentration was low in red and brown algae, but high in green seaweeds. The total content of As in relation to the group of algae followed the decreasing order: brown>red>green. The predominant form of As found in these macroalgae was organic. The difference in the accumulation of metals is given by the geographical and seasonal variability as well as from the species. Chen et al. [54] studied how Al, As, Cd, Cu, Cr, Hg, Mn, Ni, Pb, and Se were distributed in a large sample of brown and red seaweeds in an industrialized and urbanized area, also estimating the hazard risk for human health. Red algae showed higher levels of cadmium, copper, manganese, and nickel compared with brown seaweeds that contained higher concentrations of Hg and Se. Al, Cd, Cu, Hg, Mn, and Se in brown and Al, As, Cu, Mn, and Pb in red seaweeds varied significantly across different origin sites. Negligible risk for human health resulted by intake of these seaweeds. Filippini et al. [55] analyzed 72 samples bought in Italy, originating from various countries with no information on the characteristics of the geographical area of collection. Rhodophyta presented high levels of investigated elements. Brown and red seaweeds

showed, from the highest to the lowest concentration, the following order, for children and adults: As>Al>Cd>Pb. Based on US EPA (United States Environmental Protection Agency) 2007 [56] guidelines, these values were below the limits. HI value for Al must be monitored for the effects on children health. Francisco et al. [57] determined contaminant elements as Pb, Cd, Hg, and As in *Fucus spiralis* collected in coastal areas of Portugal with low levels of pollution. The As, Cd, Pb, and Hg concentration were lower than previously observed for this species in Portugal and was associated with low pollution rate of the harvesting site. Kim et al. [58] assessed heavy metals content in *Gracilaria tikvahiae*, and in *Saccharina latissima* from three sites of New York state coastal area, where there is intense anthropogenic activity. Except for Pb, the contaminant concentrations were below the governmental limits. Li et al. [59] assessed the concentrations of the polycyclic aromatic hydrocarbons (PAHs), heavy metals (arsenic, cadmium, chromium, copper, and lead), and pesticides in *U. prolifera*, in the Yellow sea, contaminated by oil spills and pollutants from the inland. The findings suggested that *U. prolifera* is safe for heavy metal levels. On the other side, PAHs concentration was remarkable. Despite this, *U. prolifera* consumption was considered to be safe, also considering the incremental lifetime cancer risk (ILCR) values, below the US EPA safety threshold. Mise et al. [60] analyzed the presence of arsenic (total and inorganic), in daily meals consumed by pregnant women and their children for three days. The results showed a high exposition to inorganic arsenic in children due to the *Hijiki* ingestion. Panebianco et al. [49], among seventy-seven commercial products, examined four types of algae from various eastern countries and from Germany to analyze heavy metal contamination from human pollution. All four seaweeds contained toxic elements. The presence of lead and arsenic was observed in two samples. Three heavy metals and six trace elements were determined by Paz et al. [61] in Phaeophyta spp. harvested in Spain, in an area where human activities, especially related to tourism, had an impact. Low dietary intake of Pb for a dose of 5 g of *P. pavonica* and *H. scoparia* was detected. The contribution to the tolerable weekly intake (TWI) suggested no health risk for 5 g of *P. pavonica* consumption regarding Al and Cd content. Paz et al. [62] also measured the concentrations of Al, Cd, Hg, and Pb, and the related toxicological risk in various edible seaweed samples sold in Spain. The seaweed derived from industrialized areas of Asian countries and from Galicia, with an intense human activity that determined environmental disasters on the coast. Al was high in the seaweed salad. Asian algae recorded high levels of Al, Cd and Pb, while high concentrations of Hg were found in algae collected in Europe. The TWI related to cadmium could pose a danger in case of daily ingestion of 5 g of dehydrated algae, but in general the safety for adults is guaranteed. Sa Monteiro et al. [63] investigated the content of cadmium, iodine, lead, mercury, and total arsenic in seaweed cultivated and harvested in Denmark. No risk was identified for all the elements considered. The study of Santos-Silva et al. [64] measured As, Cd, Cu, Hg, Pb, and Zn content in four seaweed species collected in a pollution-free area of Brazil. Brown seaweeds showed the higher As, Cd, and Hg concentrations compared to red seaweeds. Squadrone et al. [65] detected the trace elements content in the most widespread macroalgae in two protected areas and in one unprotected and populated area of Mediterranean coasts. Brown and green seaweeds showed greater bioaccumulation of metals than red algae. Stévant et al. [66] studied Cd, I, and iAs concentration in *A. esculenta* and *S. latissima* harvested in under regulated cultivation site, in France. The results exceeded the French safety limits for cadmium and iodine, while inorganic arsenic level was below the fixed threshold. Uchida et al. [46] defined heavy metal concentration in nori sauce, coming from a cultivation in a regulated area: the contamination level of cadmium and chromium did not represent a significant risk. As for arsenic, the inorganic form was below the safety threshold. The amount of histamine detected was very low. Zhang et al. [67] analyzed common and selenium-rich agri-food in samples from the Chinese markets, from unspecified sites. The results showed a high concentration of metals compared to other selenium rich agro-food samples. Seaweeds presented higher lead, arsenic, cadmium, mercury, and chromium levels. Rzymski et al. [68] studied chemical contamination in food

supplements containing *Chlorella* and *Spirulina*, from different geographical areas, mainly oriental, from unspecified sites. Rare Earths Elements (REE) content was negligible, but aluminium and lead recorded high concentration. Liu et al. [69] analyzed a common laver species belonging to the red algae phylum harvested from two Chinese coastal provinces. Ce, La, Nd, and Y (REE) concentration resulted high in all *Porphyra* samples and exceeded the limits of the law; the variability of REEs concentration and distribution was dependent on the geographic area of study. Squadrone et al. [70] assessed the possible risk for humans analyzing REE levels in in Chlorophyta, Ochrophyta, and Rhodophyta species harvested from the Northwestern Italian Sea. Seaweeds, as primary producers, had the tendency to accumulate REE, representing a source of exposure for human health. Some species of *Cladophora, Sargassum*, and *Ulva*, have been studied for the analysis of $^{210}$Po and $^{210}$Pb accumulation by Uddin et al. [71]. The isotopes' concentration was detected more in the Phaeophytes than in Chlorophytes. Miller et al. [72] analyzed some food supplements to detect anabaenopeptins, cyanopeptolins, microcystin variants, and microginins. Microcystins emerged as the peptide present in quantities higher than the fixed limit.

## 4. Discussion

Seaweeds have been used as a source of food in human diet for centuries. In Asia, algae consumption had lasted for centuries, but only recently it spreads in Europe and other Western countries [73,74]. In 2016, over thirty tons of seaweeds were produced, and almost all were cultivated, while only a small part was collected in the place of origin [75,76]. Algae are very popular among vegetarians, who use them as starters, additions and main courses [73]. In European cuisine and food production, the properties of algae as gelling, stabilizing, and thickening, have been appreciated for a long time [77]. Seaweeds are increasingly used as supplements, as in the case of *Spirulina* and *Chlorella* [74]. Algal commercial products are, commonly, laver (raw), dried, roasted, seasoned, for side dish, as soup, and as additional ingredients [78]. For the latter, algae were used in reformulating several traditional products, including pasta [79–81] (also gluten-free [82]), bread [75,83–86], vegetable soup [87], dairy products [74], fish and fish products [88], meat and meat products [75,89,90] and snacks [91]. As described in the selected papers, the forms of consumption of the algae listed above should be taken into account in the risk assessment for humans. Another important aspect is the seaweed harvesting or growing areas. In our review, particularly for the chemical risk, about 80% of the studies considered had analyzed seaweed, from natural sites, after storage and preparation for laboratory research (so in raw or semi-processed form). The remaining 20% had unknown origin and process production.

As for other novel foods, the manufacturing seems to be critical for the presence of pathogens, in the same way of what observed for insects [92,93] and duckweeds [94]. Another reason of microbiological hazard could be the site of cultivation: in outdoor tubs the risk of infestation by rodents, birds, and insects exists [95].

Comparing the results of another review on allergenic risk, it emerged that, products rich in lipids, flour algal-based, and a type of dried Chlorella were candidates for being a source of minimal allergic forms, as well as Microcystin could play a role as neurotoxin and hepatotoxin [95]. Regarding the allergenic risks, further investigations are needed to establish the potential allergenic role of the algal proteins, as Phycocyanin C, Phycobiliproteins (the main proteins of red seaweeds) and phycolectins [43].

As regards to physical risk, the consumption of algae is safe although choking events may occur in infrequent cases [47]. Besides, macroalgae consumed in the dried form could cause damage to the walls of the digestive tract, especially in fragile individuals: as reported by Schaefer and Trocinski, "the estimated annual incidence of food impaction is 13 per 100,000 and eighty percent to ninety percent occur in the distal esophagus associated with anatomic or motor abnormalities" [96]. The ingestion of processed seaweeds (dried kombu) could represent a danger in subjects with neurological problems and for young children, as well as other roasted or toasted food. Probably, by consuming algae in the

raw, softer form, this type of accident could be avoided. Severe controls and tools (as X-ray or metal detector) used during the manufacturing process could avoid consumers' health problems determined by the presence of foreign bodies [49].

The pollution of marine environments by anthropogenic activities [97] has raised concerns about the health risks associated with seaweeds consumption [54]. Heavy metal toxicity is associated with properties of the elements, such as chemical speciation and chelation, with modes of accumulation, such as dose and routes of exposure, and with individual characteristics such as age, sex and nutritional status [98]. As, Cd, Cr, Hg, and Pb (carefully monitored for their interaction with the environment) [99], possess toxicity even at minimal doses, representing a potential risk for human health [98,100]. The binding of seaweeds with heavy metals occurs through sulphated polysaccharides on cellular wall. The degree of affinity is different between the *phyla*: in fact, the brown algae, thanks to the alginates, bind more metals, while the red algae, containing agar, to a lesser extent [101,102]. The only common factor for the accumulation of heavy metals in seaweeds was the presence of human activities near the sites where the samples were collected, as confirmed by Priyadarshini et al. [103]. Comparing our work with the review by Circuncisão et al., we found the same concerns about the possibility that algae accumulate heavy metals, although this depends on many variables, as well as the lack of clear legislative limits for consumers' protection [104].

Van der Spiegel et al., in his review, was in accordance with our results for the role of the seaweeds as heavy metal accumulator, but he did not specify what elements were particularly dangerous for human health [95]. Foods containing REEs, if consumed for a long time, can become a real poison [105]. Several animal studies and epidemiological surveys reported that environmental exposure of REEs may result in harmful effects on human health at certain dosages and over a long period of time, such as nephrogenic systemic fibrosis [106] acute myocardial infarction [107], lower IQ scores in children [108], leukemia [109], malabsorption, and indigestion [110]. The amount of REE detected in the studies examined [68–70] requires attention and further investigations are needed to evaluate for the cumulative effects on human health; however, it is strongly recommended to treat thermally the seaweeds before the consumption. Another group of substances particularly monitored for their toxicity are polycyclic aromatic hydrocarbons, highly polluting products of the modern era. The health risk linked to PAHs is represented by their ability to bioaccumulate in living organisms, to diffuse in the different layers of our planet, from air to soil and water, up to animals, including humans, and persistence [111–113]. Based on the evidence emerging from a study included in our synthesis [59], there is accordance with a work of Pavoni et al. [114] to pay attention to the accumulation of PHA in the algae. Another substance that deserves further study on the potential risk to human health is a particular isotope of polonium. The $^{210}$Po isotope is an alpha emitter of the $^{238}$U series. The sources of $^{210}$Po are represented by seawaters, river discharge, and atmospheric deposition [115]. A large fraction of the total radiation exposure experienced by individuals is delivered via marine food chain transfer [116,117]. Both the isotopes were found in green macroalgae [71]. Other studies reported the presence of the isotopes considered in a brown alga too [118,119]. Our results confirmed the presence of isotopes in both classes (brown and green). For pesticides, currently under Regulation (EC) No 396/2005 of the European Parliament, a maximum residue level (MRL) for plant and animal organisms was established at the default level of 0.01 mg/kg [120]. Although our results suggest a safe consumption of pesticide content in one seaweed species [59], there are studies [95] in which the presence of pesticides in the seaweeds is reported; further investigations are needed to evaluate the effects of pesticides accumulation in the human body deriving from algae ingestion.

*Limitation of the Study*

This study is based on an extensive search in different databases by means of research strings built on some keywords, and the selection process has followed the PRISMA

methodology. Nonetheless, some relevant studies may not have been included in the investigated databases. An update of this systematic review including also other databases and gray literature could overpass this limitation.

## 5. Conclusions

Edible seaweeds are likely among the novel foods which will gain popularity in the future. However, despite the excellent nutritional properties and their extraordinary ubiquity worldwide, serious reflections are required on the potential risks linked to their consumption. The present study confirms that the risk of toxicity depends on the anthropogenic activity, the species, and the quantity of product eaten, and suggests that allergenic and physical risks for humans are not common, and controllable by means of the adoption of good hygienic practices (GHP). The focus for future studies could on climate change and its effects on the ecosystems, particularly for the marine environment where seaweed is sensitive to the change of their habitat's conditions. Moreover, it is necessary to better clarify the aspects linked to the origin of the raw products as well as the technological processes to which the seaweed are subjected. In conclusion, considering their increasing consumption, valid and shared regulations together with further investigations on the potential risks are needed in order to confirm their safety as a human food.

**Author Contributions:** Conceptualization, G.B. and C.L.; methodology, G.C. and G.G.; formal analysis, G.C. and G.G.; investigation, G.C. and G.G.; resources, G.B.; writing—original draft preparation, G.C. and G.G.; writing—review and editing, G.B., G.C., G.G., and C.L.; supervision, G.B. and C.L. All authors have read and agreed to the published version of the manuscript.

**Funding:** This research received no external funding.

**Institutional Review Board Statement:** Not applicable.

**Informed Consent Statement:** Not applicable.

**Data Availability Statement:** Not applicable.

**Conflicts of Interest:** The authors declare no conflict of interest.

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
