# Peer review of "Seaweeds as a “Palatable” Challenge between Innovation and Sustainability: A Systematic Review of Food Safety"

_sustainability, doi:10.3390/su13147652_

Round 1
Reviewer 1 Report
The review article presented for review concerns an important problem of food safety, especially novel foods, for which the human digestive tract is not adapted. Therefore, you can expect some specific microflora, chemical or allergic hazards. Therefore, the topic is important. However, in order to improve the quality of this study, it is worth considering the general and specific comments
General note.
The chemical hazards include the level of heavy metals that come from the environment. The authors did not consider the issue of natural metabolites produced by seaweed (possibly under specific conditions) that could be potentially harmful to humans.
There is also no account of the origin of the seaweed. The level of heavy metals depends on the degree of environmental pollution the seaweed comes from. So you can look for a correlation between the level of pollution and the place where the seaweed was obtained. It is also worth comparing the seaweed from biotechnological farming with those from natural reservoirs. Contamination in biotechnological cultures may be the result of the quality of the supplied medium. So again this is not a raw material problem, but an effect of a biotechnological process
In the case of processed foods containing seaweed, microbial, chemical and even allergic contamination can come from the technological process and not from the algae as a raw material. This aspect is also worth analyzing.
I do not see any justification for the analysis of physical hazards, as there is no possibility that they come from seaweed as a raw material - they are the result of a technological process.
In my opinion, the results of the research on processed products containing seaweed should be clearly distinguished from the results on seaweed as a raw material. Any threats in this group may result from the technology and production conditions. Therefore, it would require an analysis of the production process itself.
Detailed comments
Fig 1, 2, 3, 4 - their quality should be improved
4.1 .. Limitation (..) - remove one period
Line 402-3. The authors express the opinion that the strings used limited the scope of the search for plastics. However, the strings used for the search were created by the authors and did not prevent this aspect from being included in the search. If this has not been done consciously, it cannot be considered a limitation.
Conclusion
In conclusion, the authors state that "... the risk of toxicity depends on the anthropogenic activity, the species, and the quantity of product eaten". However, the study lacks an analysis of the dependence of toxicity on the species or the amount of consumed product.
Author Response
Dear Editor,
Thank you for having considered our article.
We submit our revised manuscript entitled “Seaweeds as a "palatable" challenge between innovation and sustainability: a systematic review of food safety” to be considered for publication on Sustainability.
All the authors have revised the work based on the suggestions and the observations of reviewers.
All the modifications have been reported in red.
Please see the attachment.
We appreciate your time and look forward to your response.
Kind regards
Giuseppe Cavallo

Reviewer 2 Report
Edible seaweeds are likely among the novel foods which will gain popularity in the future. But, despite the excellent nutritional properties and their extraordinary ubiquity worldwide, serious reflections are required on the potential risks linked to their consumption. The purpose of this systematic review was to assess microbiological, chemical, physical, and allergenic risks associated with seaweed consumption. Four research strings have been used to search for these risks. PRISMA (Preferred Reporting Item for Systematic Reviews and Meta-analysis) guidelines were applied. Finally, 39 articles met the selected criteria. No significant hazards for microbiological, allergenic, and physical risks were detected. The study confirms that the risk of toxicity depends on the anthropogenic activity, the species, and the quantity of product eaten and suggests that allergenic and physical risks for humans are not common and controllable by means of the adoption of good hygienic practices (GHP). The paper is well done but I have some remarks: - the authors should improve the figures 1-2-3-4 - the authors should extender the Conclusions. They are short and should explain better about the development of the research for the futureAuthor Response
Dear Editor,
Thank you for having considered our article.
We submit our revised manuscript entitled “Seaweeds as a "palatable" challenge between innovation and sustainability: a systematic review of food safety” to be considered for publication on Sustainability.
All the authors have revised the work based on the suggestions and the observations of reviewers.
All the modifications have been reported in red. Please see the attachment.
We appreciate your time and look forward to your response.
Kind regards Giuseppe Cavallo Department of Health Science University of Florence Viale Morgagni,48 50134 Firenze – Italy

Reviewer 3 Report
I thank you for the opportunity to review this systematic review of food safety. The topic is timely and the literature on food problems especially global food shortages has burgeoned in recent years. I agree, that ‘a novel foods, like the seaweeds, have the potential to increase food yields so that to contribute in preventing or avoiding future global food shortages’. In my opinion, in its current stage, the manuscript is well-developed, clear, and convincing enough to justify publication.
Please see my detailed comments below.
The structure of the work is an example of developing the subject of interest by the Author/s. The topic is highly relevant and topical given the accelerating quantum of change facing today’s world. I found the article well-written and cohesive. The title captures the importance of the study. The abstract is complete; essential details are presented. The research is based on a sound literature review and well-considered questions arising from this literature review. The theoretical framework is well researched and discussed in detail. The research design is defined and clearly described, and is sufficiently detailed to permit the study to replicated. The use of PRISMA methodology in this type of research provides very interesting findings and proves the reliability of the analysis. The design and conduct of the study are plausible. Data analysis procedures are also sufficiently described and are sufficiently detailed to permit the study to be replicated. I was happy to read about the results and conclusions.
Results are organized in a way that is easy to understand. The conclusions are clearly stated; key points stand out. The study adds new value to the literature already available on the subject. In my opinion, the article is suitable for the publication on Sustainability journal.
Once again - congratulations. I encourage the Author/s to further research in this area.
Author Response
Dear Editor,
Thank you for having considered our article.
We submit our revised manuscript entitled “Seaweeds as a "palatable" challenge between innovation and sustainability: a systematic review of food safety” to be considered for publication on Sustainability.
All the authors have revised the work based on the suggestions and the observations of reviewers.
All the modifications have been reported in red. Please see the attachment.
We appreciate your time and look forward to your response.
Kind regards Giuseppe Cavallo Department of Health Science University of Florence Viale Morgagni,48 50134 Firenze – Italy

Round 2
Reviewer 1 Report
The authors took my suggestions into account. So, I recommend the article to be printed in its current form
Author Response
Thanks for your evaluation.
Best regards
This manuscript is a resubmission of an earlier submission. The following is a list of the peer review reports and author responses from that submission.